# Early cerebral hypoxia in extremely preterm infants and neurodevelopmental impairment at 2 year of age: A post hoc analysis of the SafeBoosC II trial

**Anne Mette Plomgaard**[1]*, **Christoph E. Schwarz**[2,3], **Olivier Claris**[4], **Eugene M. Dempsey**[3], **Monica Fumagalli**[5,6], **Simon Hyttel-Sorensen**[1], **Petra Lemmers**[7], **Adelina Pellicer**[8], **Gerhard Pichler**[9], **Gorm Greisen**[1]

**1** Department of Neonatology, Rigshospitalet, Copenhagen University Hospital, Copenhagen, Denmark, **2** Department of Neonatology, University Children's Hospital, Tuebingen, Germany, **3** INFANT Centre, University College Cork, Cork, Ireland, **4** Department of Neonatology, Hospices Civils de Lyon, Claude Bernard University, Lyon, France, **5** NICU, Fondazione IRCCS Ca' Granda Ospedale Maggiore Policlinico, Milan, Italy, **6** Department of Clinical Sciences and Community Health, University of Milan, Milan, Italy, **7** Wilhelmina Children's Hospital, University Medical Center Utrecht, Utrecht, The Netherlands, **8** Department of Neonatology, La Paz University Hospital, Madrid, Spain, **9** Department of Pediatrics, Medical University of Graz, Graz, Austria

* amplomgaard@gmail.com

## Abstract

### Background

The SafeBoosC II, randomised clinical trial, showed that the burden of cerebral hypoxia was reduced with the combination of near infrared spectroscopy and a treatment guideline in extremely preterm infants during the first 72 hours after birth. We have previously reported that a high burden of cerebral hypoxia was associated with cerebral haemorrhage and EEG suppression towards the end of the 72-hour intervention period, regardless of allocation. In this study we describe the associations between the burden of cerebral hypoxia and the 2-year outcome.

### Methods

Cerebral oxygenation was continuously monitored from 3 to 72 hours after birth in 166 extremely preterm infants. At 2 years of age 114 of 133 surviving children participated in the follow-up program: medical examination, Bayley II or III test and the parental Ages and Stages Questionnaire. The infants were classified according to the burden of hypoxia: within the first three quartiles (n = 86, low burden) or within in the 4th quartile (n = 28, high burden). All analyses were conducted post hoc.

### Results

There were no statistically significant differences between the quantitative assessments of neurodevelopment in the groups of infants with the low burden of cerebral hypoxia versus the group of infants with the high burden of cerebral hypoxia. The infants in the high hypoxia

**Data Availability Statement:** This is a small dataset on an easily identified population at named institutions during an identified time period. While

we did make the data on the original trial available to researchers, now when the data includes the two-year follow-up, including data on obvious motor disabilities in a subset of children, we find it has greater privacy concerns, and therefore will not share the data. Data Availability: The data is sensitive patient data and is stored and analysed under the restrictions of the Danish Data Protection Agency. The dataset used for the present analysis can be obtained from the authors at request for research purposes. Note that it will be anonymized, without NICU identity, with dicotomised birth weight and gestational age data to limit the risk of re-identification, and on condition of not disclosing data to third parties, but referring requests of access back to us. Please contact The Capital Region, Videnscenter for dataanmeldelser by mail (cru-fp-vfd@regionh.dk).

**Funding:** GG was the principal investigator on the SafeBoosC-II trial. He was funded in Jan 2012 by The Danish Council for Strategic Research with 11.1 mio dkk. https://ufm.dk/aktuelt/nyheder/2011/bevillinger-fra-det-strategiske-forskningsrad-programkomiteen-for-individ-sygdom-og-samfund-november-2011 The funder had no role in study design, data collection and analysis, decision to publish, or preparation of the manuscript.

**Competing interests:** The authors have declared that no competing interests exist.

**Abbreviations:** ASQ, Ages&Stages Questionnaire; Bayley II/III, Bayley Scales of Infant and Toddler Development, Second or Third Edition; cUS, cranial ultrasound; GMFCS, Gross Motor Function Classification System; MDI, Mental Development Index; NIRS, Near infrared spectroscopy; SafeBoosC II trial, Safeguarding the Brains of our smallest Children, a phase II feasibility multicentre randomised clinical trial; rStO$_2$, regional oxygen saturation.

burden group had a higher–though again not statistically significant—rate of cerebral palsy (OR 2.14 (0.33–13.78)) and severe developmental impairment (OR 4.74 (0.74–30.49)).

## Conclusions

The burden of cerebral hypoxia was not significantly associated with impaired 2-year neuro-developmental outcome in this post-hoc analysis of a feasibility trial.

## Introduction

Being born extremely preterm is associated with a risk of death and later moderate- to severe neurodevelopmental impairment. Several studies have shown associations between cerebral hypoxia, as measured by cerebral near infrared spectroscopy (NIRS), and neurodevelopmental impairment [1–4]. NIRS-guided interventions have been shown to reduce the cerebral burden of hypoxia both in the first 15 minutes after birth in preterm infants [5] and during the first 72 hours of life in extremely preterm infants by [6]. NIRS-guided management, however, did not improve early biomarkers of brain injury [7] or outcome at 2 years [8]. A post-hoc analysis showed that a burden of cerebral hypoxia in the highest quartile was significantly associated with death before discharge, severe brain injury on cranial ultrasound (cUS), and reduced electrical brain activity compared to infants with the burden of cerebral hypoxia in the three lower quartiles in the SafeBoosC-II trial, regardless of group allocation [9]. In this post-hoc analysis we wish to explore the relationship between the burden of cerebral hypoxia during the first 72 hours of life and developmental outcome at two years corrected age in the extremely preterm infants included in the SafeBoosC-II trial while disregarding group allocation.

## Patients and methods

The SafeBoosC II trial was a multicentre randomised trial with eight centres recruiting a total of 166 preterm infants within 3 hours after birth. The study was conducted between June 2012 and December 2013 [6]. Of these 133 infants were alive at discharge and 114 children were included in the follow up analysis. The burden of cerebral hypoxia did not differ between the infants who were followed up and the infants who were not [8]. The trial was registered at ClinicalTrial.gov, NCT01590316, the protocol is published [10] and available in full at http://www.safeboosc.eu.

### The burden of cerebral tissue hypoxia

The infants were included in the SafeBoosC II study three hours after birth and regional cerebral oxygenation (rStO$_2$) was continuously monitored by NIRS during until 72 hours after birth. The infants were randomised to the intervention or control group. Intervention group: the cerebral oxygenation level was visible and the target range was set at 55% to 85%. The clinicians were provided with a dedicated treatment guideline listing possible interventions if the cerebral oxygenation was out of range [11]. Control group: the cerebral oxygenation levels were recorded, but the clinicians were blinded to the results and the infants were given standard treatment and care, only. The primary outcome of the SafeBoosC II trial was the combined burden of hypo- and hyperoxia, calculated as time spent outside the target range of rStO$_2$ predefined as 55–85%. The burden of cerebral hypoxia was calculated as the time spent below the target limits multiplied by the mean deviation from the lower limit during the first

72 hours of life, expressed as percentage hours (%hours. e.g. 10%hours represent one hour with 45% $rStO_2$ or 2 hours with a 50% value of $rStO_2$).

In this post-hoc analysis we wish to explore the relationship between the burden of cerebral hypoxia and developmental outcome at two years corrected age, regardless of trial group allocation.

## Cranial ultrasound

cUS was performed at day 1, 4, 7, 14, and 35 and at term equivalent age. The cUS was categorised as no, mild/moderate or severe brain injury as previously described [12].

No brain injury: None of the findings below.

Mild/moderate brain injury: Grade 1–2 Intraventricular haemorrhage (IVH, including germinal layer haemorrhage), isolated ventriculomegaly with ventricular index <p97a, inhomogeneous flaring persisting after day 7, or global thinning of corpus callosum at term equivalent age.

Severe brain injury: IVH III (ventricular index >p97 during the acute phase), post haemorrhagic ventricular dilatation, parenchymal/periventricular haemorrhagic infarction, unilateral porencephalic cysts, cystic periventricular leukomalacia (bilateral), cerebellar haemorrhage, cerebral atrophy at term age or stroke.

## Neurodevelopmental evaluation

At 2 years corrected age, the participants were invited to a follow-up visit consisting of a medical examination and an assessment of their neurodevelopment with the Bayley II or III, and the parents were asked to fill in an Ages and Stages Questionnaire (ASQ) [8].

Medical examination: Basic growth measurements were collected. Vision and auditory functions were evaluated. If the child showed signs of cerebral palsy, the gross motor function was classified using the Gross Motor Function Classification System (GMFCS). The doctor performing the medical examination may not have been blind to the intervention.

Bayley Scales of Infant and Toddler Development, Second Edition (Bayley- II) or Third Edition (Bayley-III), depending on what version the centre was using at the time of the study. Bayley III is known to underestimate the developmental deficit when compared to Bayley II [13], we therefore calculated the predicted mental developmental index for the Bayley III edition, as previously described [14]. The psychologists conducting the tests were blinded to cerebral NIRS outcome and imaging results.

ASQ: A set of parental questionnaires covering the development of children aged from four months to five years. Each questionnaire contains five developmental domains, communications, gross motor skills, fine motor skills, problem-solving and personal-social skills. The parents returned the questionnaires by mail or handed them in on the day of medical examination. The parents were not blinded to intervention group nor imaging results.

Neurodevelopmental impairment was classified according to Classification of health status at 2 years as a perinatal outcome, British Association of Perinatal Medicine [15]. Severe neurodevelopmental disability if any of the following conditions was present: Cerebral palsy (CP) with a GMFCS score of 3–5; a cognitive function score below -3 standard deviations scores, Mental Development Index (MDI) below 55; hearing impairment with no useful hearing even with aids; no meaningful words or signs and blind or only able to perceive light or light reflecting objects. Moderate neurodevelopmental disability if any of the following conditions was present: CP with a GMFCS score of 1–2; a cognitive function score between -3 and -2 standard deviations scores, (MDI 55–70); hearing impairment, but useful hearing with aids; fewer than five words or signs and moderately impaired vision.

## Statistics

The goal of the statistical analysis was to examine the association between neonatal cerebral hypoxia and neurodevelopmental outcome. For this purpose, the median and interquartile range was determined for the burden of cerebral hypoxia and the infants were divided in two groups according to a burden within the 4th quartile (highest burden of cerebral hypoxia) or below (lower 3 quartiles of burden of cerebral hypoxia). This dichotomisation was previous used in our group when comparing biomarkers [9] and chosen a priory for the present post-hoc data analysis.

We compared the baseline patient characteristics, the interventions conducted during the 72 hours NIRS monitoring, the neonatal complications, and the outcomes between the two groups, using the t-test and Chi-square test as appropriate. Thereafter we calculated the mean difference and the 95% confidence intervals for the ASQ score and Bayley II/III scores and the odds ratio and 95% confidence interval for adverse outcome in the group of infants with in the 4th quartile versus the children in the three lower quartiles.

If baseline patient characteristics, the interventions conducted during the 72 hours NIRS monitoring, or the neonatal morbidity rate (severe brain injury on cUS, necrotising enterocolitis, bronchopulmonary dysplasia or retinopathy of prematurity) differed significantly ($p < 0.05$) and the differences between the developmental outcomes in the two groups were significant ($p < 0.05$), additional multiple logistic and linear regression analyses was planned–as the basis for discussing the possibilities as regards causal paths as well as confounding.

None of the analyses reported here were specified in the SafeBoosC-II study protocol.

The statistics was performed using IBM SPSS Statistics for Windows Version 20.0 (Armonk, New York, USA).

## Ethics

The SafeBoosC II trial was approved by each hospital's research ethics committee: Ethikkommission, Medizinische Universität Graz, Austria; De videnskabetiske komiteer, Copenhagen, Denmark; Comité de Protection des Personnes Sud Est III–Groupement Hospitalier Edouard Herriot, Lyon, France; Clinical Research Ethics Committee, University College Cork, Ireland; Comitato de Etica, Drezione Scientifica, Fondazione IRCCSA Ca Granda, Ospetale Maggiore Plicliico, Milano, Italy; De Medische Ethische Toetsingescommissie, Universitair Medisch Centum, Utrecht, Netherland; Comité Etico de Investigación Clinica, Hospital Universitario La Paz, Madrid, Spain, and where required also by the competent authority responsible for medical devices (Austria, Denmark and France). Written informed parental consent was mandatory before inclusion. The trial was conducted according to the guidelines of the Declaration of Helsinki and the International Conference on Harmonisation good clinical practice.

## Results

One-hundred sixty-six children were randomised to the SafeBoosC-II trial. Due missing data on cerebral oxygenation, death before discharge, death after discharge and loss to follow-up only data of 114 children were included in this post-hoc follow up analysis (**Fig 1**). The children were classified into two groups according to the burden of cerebral hypoxia: children within the first three quartiles (n = 86, low burden) and children within in the 4th quartile (n = 28, high burden). The ranges of hypoxia in the 2 groups were: 0.1–78.3%hours in the low group and 78.7 to 803.9%hours in the high group.

The baseline patient characteristics (**Table 1**) and treatment and major neonatal morbidities (**Table 2**) were compared. There were significantly fewer boys in the high group, more

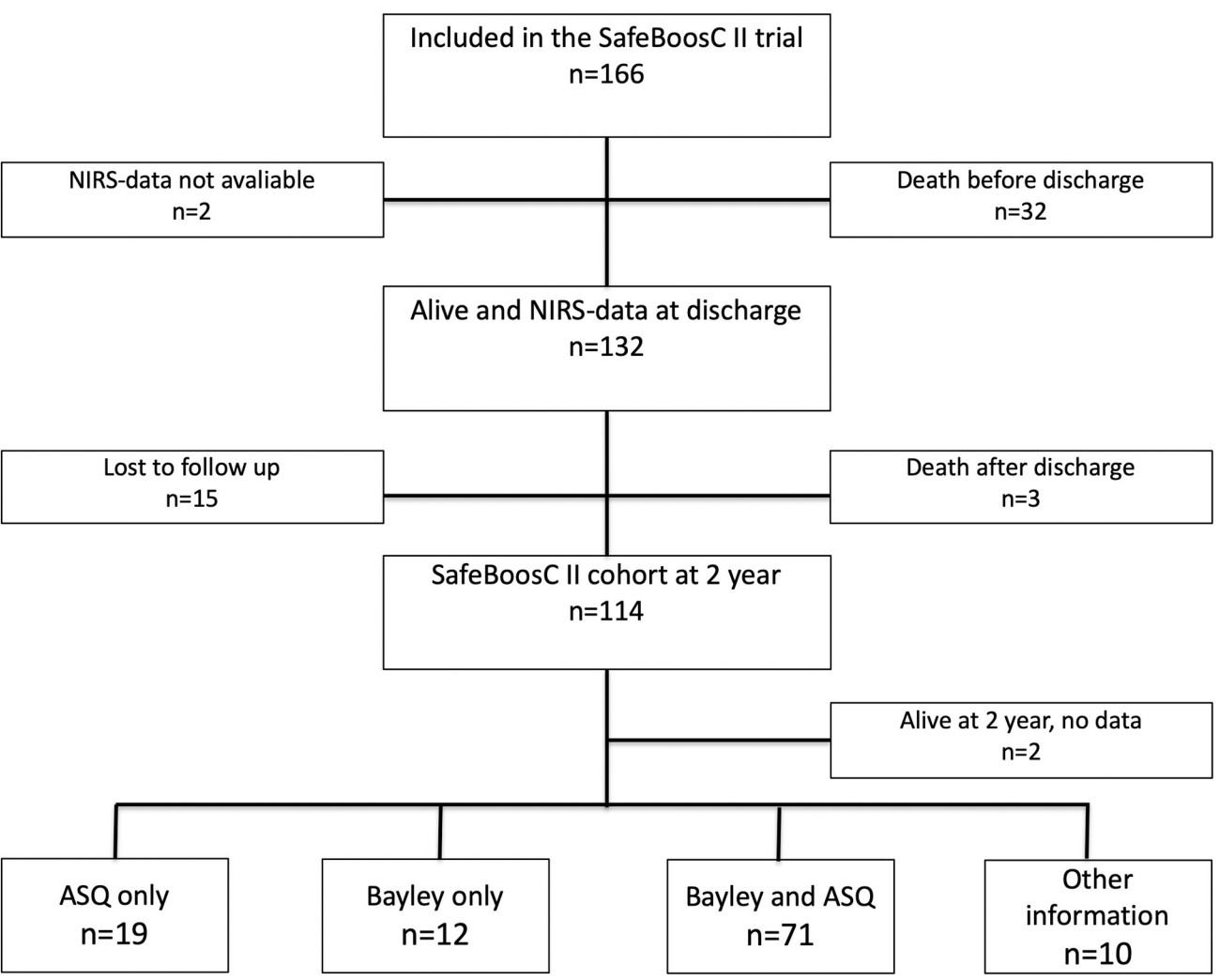

**Fig 1. Flowchart from inclusion in the randomised trial to the two-year neurodevelopmental outcomes.**

**Table 1. Baseline patient characteristics.**

| | Burden of cerebral hypoxia | | |
|---|---|---|---|
| | Quartile 1 to 3 | Quartile 4 | p-value |
| | n = 86 | n = 28 | |
| Gestational age (week), mean (SD) | 26.6 (1.1) | 26.7 (1.2) | 0.72 |
| Birth weight (gram), mean (SD) | 878 (212) | 892 (191) | 0.71 |
| Male sex, n (%) | 46 (53) | 8 (29) | 0.04 |
| Twins, n (%) | 16 (19) | 10 (36) | 0.11 |
| Antenatal steroids full course, n (%) | 61 (71) | 25 (89) | 0.13 |
| Prolonged rupture of membranes, n (%) | 26 (30) | 11 (41) | 0.44 |
| Maternal chorioamnionitis, n (%) | 3 (4) | 3 (11) | 0.33 |
| Umbilical pH, mean (SD) | 7.31 (0.09) | 7.31 (0.08) | 0.69 |
| APGAR score <5 points at 5 minutes, n (%) | 13 (15) | 7 (25) | 0.36 |

Between group comparison was performed using the t-test and Chi-square test as appropriate.

**Table 2. Treatment during the first 72 hours and major neonatal morbidities.**

| | Burden of cerebral hypoxia | | |
| --- | --- | --- | --- |
| | Quartile 1 to 3 | Quartile 4 | p-value |
| **Treatment during the first 72h of life** | **n = 86** | **n = 28** | |
| Surfactant treatment, n (%) | 62 (72) | 19 (69) | 0.67 |
| Mechanical ventilation, n (%) | 52 (60) | 15 (54) | 0.52 |
| Persistent ductus arteriosus treatment, n (%) | 10 (12) | 2 (7) | 0.73 |
| Use of vasopressors/inotropes, n (%) | 9 (11) | 10 (37) | 0.01 |
| Any red blood cell transfusion, n (%) | 20 (23) | 10 (37) | 0.16 |
| Corticosteroids, n (%) | 0 (0) | 2 (7) | 0.06 |
| **Intervention group, n (%)** | 55 (64) | 10 (37) | 0.01 |
| **Major neonatal morbidities** | | | |
| Severe brain injury on cranial ultrasound, n (%) | 9 (10) | 6 (21) | 0.19 |
| Necrotising enterocolitis, n (%) | 8 (9) | 4 (14) | 0.49 |
| Bronchopulmonary dysplasia, n (%) | 46 (54) | 14 (50) | 0.71 |
| Retinopathy of prematurity, n (%) | 12 (14) | 5 (18) | 0.76 |

Between group comparison was performed using the t-test and Chi-square test as appropriate.

children received vasopressors or inotropes. There were also more control group infants from the randomised trial in the high burden group versus low burden group.

The rates of major neonatal morbidities complication rates were similar (Table 2).

There were no statistically significant differences between the Bayley II or III sub- or total scores when comparing the group of infants form the high burden of cerebral hypoxia versus the low burden group (Table 3). The combined outcome of Bayley II and III for the 2 groups was mean (SD) 83.6 (17.6) for the children with a high burden of cerebral hypoxia versus 90.5 (16.9) for the children with the low burden of cerebral hypoxia (Cohen's d = -0.40, 95% CI: -0.12 to 0.93, p = 0.13). There was no clear visible relation between developmental scores and the burden of cerebral hypoxia (Fig 2). The ASQ-scores were similar in the two groups (Table 1). The rates of developmental impairment were higher in the high hypoxic burden group: cerebral palsy (OR 2.14 (0.33–13.78)) and severe developmental impairment (OR 4.74 (0.74–30.49) (Table 3), but these differences were not statistically significant (p = 0.59 and 0.11).

Multiple regression was not done, since no statistically significant effects of cerebral hypoxia was found on bivariate analysis.

## Discussion

In this post–hoc analysis of data collected as part of a randomised controlled phase II trial we found no statistically significant differences at two years of age between the children in the high hypoxic burden group compared to the low hypoxic burden group.

The SafeBoosC II was a randomised trial with the primary objective of investigating the possibility of reducing the combined burden of cerebral hyper- and hypoxia. As this showed possible, more infants from the control group ended up in the high burden group used in the present analysis. We focused solely on the hypoxic burden since the burden of cerebral hyperoxia was negligible and unaffected by the intervention [6].

Several publications report that low cerebral oxygenation is associated with developmental impairment of preterm infants. Among 67 infants born below 32 weeks of gestation, those within the lower (as well as higher) quartile of $rStO_2$ in the neonatal period more often had impaired cognitive outcome at 2–3 year [4]. The burden of tissue hypoxia was defined as time

**Table 3. Developmental outcome at 2 year of age.**

| | Burden of cerebral hypoxia | | | |
|---|---|---|---|---|
| | Quartile 1 to 3 | Quartile 4 | p-value | Mean difference (95% CI) |
| **Bayley II (n)** | **22** | **8** | | |
| Age (months), mean (SE) | 24.8 (0.3) | 25.2 (0.3) | | |
| Psychomotor developmental index, mean (SE) | 87.6 (3.0) | 81.1 (3.9) | | |
| Mental developmental index, mean (SE) | 93.5 (3.1) | 85.3 (3.4) | 0.18 | -8.2 (-20.4 to +3.9) |
| **Bayley III (n)** | **39** | **13** | | |
| Age (months), mean (SE) | 24.6 (0.6) | 24.7 (0.5) | | |
| Cognitive index, mean (SE) | 98.1 (2.6) | 91.3 (4.8) | | |
| Language index, mean (SE) | 95.4 (2.8) | 91.6 (4.0) | | |
| Motor index, mean (SE) | 95.4 (2.4) | 93.1 (3.7) | | |
| Predicted mental developmental index, mean (SE) | 88.6 (3.0) | 82.5 (5.7) | 0.35 | -6.2 (-19.3 to +7.0) |
| **Ages and Stages Questionnaire (ASQ) (n)** | **69** | **21** | | |
| Age (months) mean (SE) | 25.8 (0.5) | 25.2 (0.4) | | |
| Gross motor score, mean (SE) | 42.4 (2.0) | 40.5 (3.5) | | |
| Fine motor score, mean (SE) | 43.6 (2.0) | 42.8 (2.4) | | |
| Communication score, mean (SE) | 43.5 (2.0) | 43.8 (3.3) | | |
| Personal social score, mean (SE) | 42.5 (1.7) | 42.4 (2.9) | | |
| Problem solving score, mean (SE) | 43.8 (1.7) | 41.4 (2.7) | | |
| Total ASQ score, mean (SE) | 215.5 (7.4) | 210.9 (12.4) | 0.76 | -4.6 (-31.5 to +23.3) |
| **Medical examination (n)** | **63** | **21** | | **Odds ratio (95% diff)** |
| Head circumference (centimetre), mean (SE) | 48.1 (0.2) | 48.0 (0.4) | | |
| Weight (kg), mean (SE) | 11.1 (0.2) | 11.2 (0.3) | | |
| Height (centimetre), mean (SE) | 85.2 (0.6) | 85.5 (0.9) | | |
| Cerebral palsy, n (%) | 3 (5) | 2 (10) | 0.59 | 2.14 (0.33–13.78) |
| Vision impairment, n (%) | 2 (3) | 2 (10) | 0.28 | 3.21 (0.43–24.36) |
| **Combined outcomes (n)** | **62** | **22** | | |
| Moderate neurodevelopmental impairment, n (%) | 8 (13) | 3 (14) | 0.93 | 1.07 (0.26–4.34) |
| Severe neurodevelopmental impairment, n (%) | 2 (3) | 3 (14) | 0.11 | 4.74 (0.74–30.49) |
| Moderate or severe developmental impairment, n (%) | 10 (16) | 6 (27) | 0.25 | 1.95 (0.61–6.02) |

Between group comparison was performed using the t-test and Chi-square test as appropriate.

with rStO$_2$ below 50% and cerebral oxygenation was estimated from a 2-hour period of recording each day, while in the infants in the SafeBoosC II trial, the threshold was 55% and cerebral tissue oxygenation was continuously monitored from 3 hours after birth until 72 hours after birth [10]. In a cohort of 724 infants born at gestational age below 32 weeks and monitored with cerebral NIRS during the first 72 hours after birth, spending more than 20% of the time with rStO$_2$ below 55% was associated with impaired cognitive outcome at 2 years corrected age (Odds Ratio 1.4 95% CI 1.1–1.7) [1]. The same group published that prolonged suboptimal rStO$_2$ in preterm infants with patent ductus arteriosus was associated with reduced cerebral brain volume at term equivalent age [3]. They also identified that regardless of mean arterial blood pressure cerebral oxygenation below 50% for more than 10% of the time was related to adverse outcome at 18 months in 66 preterm infants born at gestational age below 32 weeks [2].

Several other studies have shown an association between cerebral oxygenation immediately after birth [16] or within the first 3 days [17] with adverse short-term outcome in preterm infants.

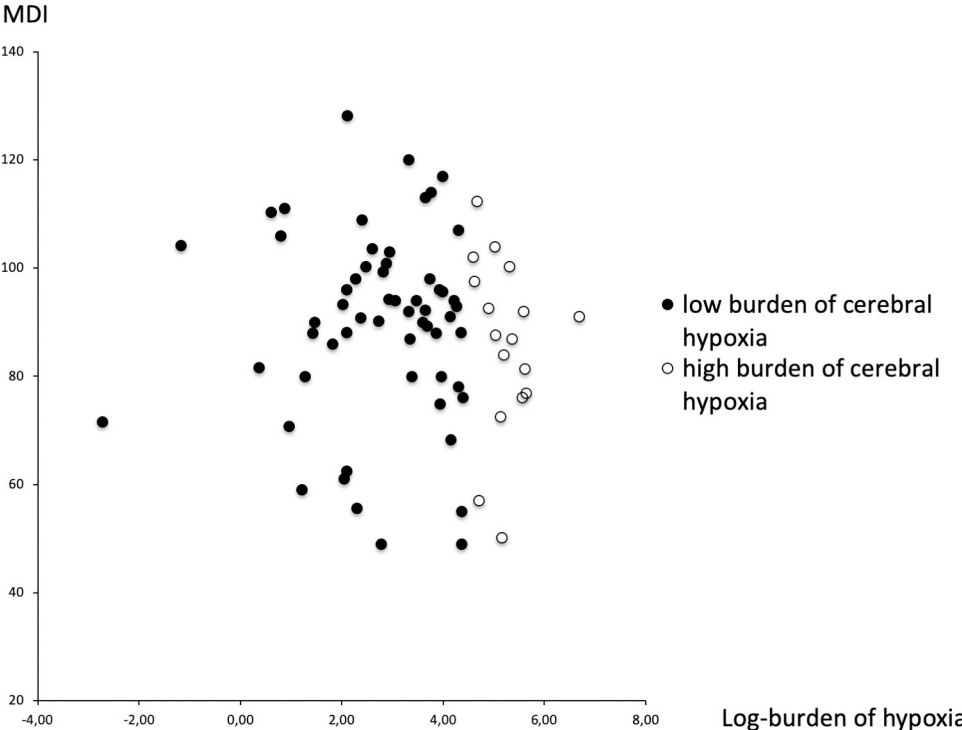

**Fig 2. Burden of cerebral hypoxia and mental development index at 2 year.** Log-burden of hypoxia and Mental Development Index (MDI) combined of Bayley II and III at 2 year corrected age. The X-axis is a logarithmic scale (base2), i.e. one unit corresponds to a doubling of the hypoxic burden, and the limit between the two groups is at 78% hours. There is no suggestion of a different threshold of hypoxic burden, nor of a correlation across the scales.

We previously reported that death, brain injury, and reduced brain electrical activity were more common in the high hypoxia burden group compared to the low burden group, i.e. precisely the groups that were also compared in the present analysis [9]. We were not able to demonstrate a similar association to neurodevelopmental deficits at two years of age. This may just be due to the limited statistical power, thus the confidence limits included clinical relevant associations. On the other hand, it is possible that the 'true' effects of cerebral hypoxia 'wane' with age due to the multifactorial aetiology of long-term outcome in preterm infants. In particular, it should be mentioned that severe brain injury as defined by cranial ultrasound imaging in the neonatal period is only a moderately strong predictor of moderate-to-severe neurodevelopmental deficit at two years of age or later.

### Strength and limitations

The strengths are the use of data from a randomised controlled trial with prospective and well-defined data collection, as well as a pre-defined classification of the hypoxic exposure. In this way the problem with multiple results of exploratory, post-hoc analyses was minimised.

Our sample size was relatively small: 114 extremely preterm infants and the incidence of moderate-to-severe developmental impairment was relatively low (16 of 114 infants (14%). Thus, the statistical strength with this sample size is limited and the confidence limits are wide and include clinically relevant effect sizes. Furthermore, the original approach of the SafeBoosC-II trial, using a simple area-under-the curve to quantify the burden of hypoxia may be too simple for the purpose of associating with brain injury and neurodevelopmental consequences. For pulse oximetry, the duration of the individual episodes seems to be important in particular

[18, 19]. Furthermore, more infants in the high hypoxia burden group received vasopressors or inotropes and red blood cell transfusions, so they likely were more sick than the infants in the low burden group, on the other hand this group included fewer boys who are at increased risk of neurodevelopmental impairment. Since the differences between the two groups in terms of neurodevelopmental impairment were not statistically significant, we abstained from additional analyses to adjust for potential confounding. In any case, causal inference would be difficult given the uncertain direction of effect among the neonatal variables. As a further weakness, the cerebral oxygenation reported by the two different devices used in the trial is not identical and using a single cut-off value may not be optimal [20–22].

Finally, the SafeBoosC II study allowed the use of either Bayley II or III and in order to compare the two editions the results from the Bayley III was converted into a predicted MDI [14]–as this procedure was conducted in both groups, it should not affect the result. However developmental delay as measured by Bayley at 2 years of age may not be the best predictor later cognitive outcomes [23].

## Conclusion

The burden of cerebral tissue hypoxia was not statistically significantly associated with 2-year neurodevelopmental outcome in this post-hoc analysis of data from a randomised controlled trial, but the confidence limits were wide, and a clinically relevant association could not be excluded. A larger, appropriately powered randomised trial is needed to test the clinical value of cerebral oximetry in preterm infants.

## Author Contributions

**Conceptualization:** Anne Mette Plomgaard, Olivier Claris, Eugene M. Dempsey, Monica Fumagalli, Simon Hyttel-Sorensen, Petra Lemmers, Adelina Pellicer, Gerhard Pichler, Gorm Greisen.

**Data curation:** Anne Mette Plomgaard, Olivier Claris, Eugene M. Dempsey, Monica Fumagalli, Simon Hyttel-Sorensen, Petra Lemmers, Adelina Pellicer, Gerhard Pichler, Gorm Greisen.

**Formal analysis:** Anne Mette Plomgaard, Christoph E. Schwarz, Simon Hyttel-Sorensen, Gorm Greisen.

**Funding acquisition:** Eugene M. Dempsey, Monica Fumagalli, Adelina Pellicer, Gerhard Pichler, Gorm Greisen.

**Investigation:** Anne Mette Plomgaard, Olivier Claris, Eugene M. Dempsey, Monica Fumagalli, Simon Hyttel-Sorensen, Petra Lemmers, Adelina Pellicer, Gerhard Pichler, Gorm Greisen.

**Methodology:** Anne Mette Plomgaard, Christoph E. Schwarz, Monica Fumagalli, Simon Hyttel-Sorensen, Petra Lemmers, Adelina Pellicer, Gerhard Pichler, Gorm Greisen.

**Project administration:** Anne Mette Plomgaard, Simon Hyttel-Sorensen, Gorm Greisen.

**Resources:** Christoph E. Schwarz, Olivier Claris, Eugene M. Dempsey.

**Validation:** Christoph E. Schwarz.

**Writing – original draft:** Anne Mette Plomgaard, Christoph E. Schwarz, Gorm Greisen.

**Writing – review & editing:** Olivier Claris, Eugene M. Dempsey, Monica Fumagalli, Simon Hyttel-Sorensen, Petra Lemmers, Adelina Pellicer, Gerhard Pichler.

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
