## [Decision Letter · Decision Letter 0]

7 Oct 2020

PONE-D-20-27390

Early cerebral hypoxia in extremely preterm infants and neurodevelopmental impairment at 2 year of age: A Post hoc analysis of the SafeBoosC II trial

PLOS ONE

Dear Anne Mette Plomgaard,

Thank you for submitting your manuscript to PLOS ONE. After careful consideration, we feel that it has merit but does not fully meet PLOS ONE’s publication criteria as it currently stands. Therefore, we invite you to submit a revised version of the manuscript that addresses the points raised during the review process.

We look forward to receiving your revised manuscript.

Kind regards,

Georg M. Schmölzer

Academic Editor

PLOS ONE

Journal Requirements:

2. Thank you for including your ethics statement: 'The SafeBoosC II trial was approved by each hospital’s research ethics committee and where required also by the competent authority responsible for medical devices. Written informed parental consent was mandatory before inclusion. The trial was conducted according to the guidelines of the Declaration of Helsinki and the International Conference on Harmonisation good clinical practice.'

3. For more information on PLOS ONE's expectations for statistical reporting, please see https://journals.plos.org/plosone/s/submission-guidelines.#loc-statistical-reporting. Please update your Methods and Results sections accordingly.

4. We noted in your submission details that a portion of your manuscript may have been presented or published elsewhere.

[Yes.

This manuscript contains data previously published. We clearly state in this manuscript that the analyses are conducted  this post-hoc -and was not planned per protocol.

I attach related manuscript.]

5.We note that you have indicated that data from this study are available upon request. PLOS only allows data to be available upon request if there are legal or ethical restrictions on sharing data publicly. For information on unacceptable data access restrictions, please see http://journals.plos.org/plosone/s/data-availability#loc-unacceptable-data-access-restrictions.

Reviewers' comments:

Reviewer's Responses to Questions

**Comments to the Author**

1. Is the manuscript technically sound, and do the data support the conclusions?

Reviewer #1: Partly

Reviewer #2: Partly

2. Has the statistical analysis been performed appropriately and rigorously? 

Reviewer #1: Yes

Reviewer #2: No

3. Have the authors made all data underlying the findings in their manuscript fully available?

Reviewer #1: Yes

Reviewer #2: No

4. Is the manuscript presented in an intelligible fashion and written in standard English?

Reviewer #1: Yes

Reviewer #2: Yes

5. Review Comments to the Author

Reviewer #1: Early cerebral hypoxia in extremely preterm infants and neurodevelopmental impairment at 2 years of age: A post hoc analysis of the SafeBoosc II trial (ReviewPLOSone-2020-1005)

This report is an exploratory post-hoc analysis of the SafeBoosC II trial to assess if there is a relationship between the burden of cerebral hypoxia and neurodevelopmental outcome at 2 years corrected age. The SafeBoosC II trial was a randomized study among 166 preterm infants to determine if continuous NIRS recording of cerebral oxygenation over the first 72 hours after birth combined with a protocol to maintain cerebral oxygenation between 55-85% reduces the time with cerebral hypoxia or hyperoxia. A series of publications have demonstrated that the intervention could reduce the time with cerebral hypoxia but outcome measures (biomarkers, EEG, imaging, neurodevelopment) have not be altered. The current manuscript builds on an earlier publication reporting that infants in the highest quartile of cerebral hypoxia exposure had more severe intracranial hemorrhage, altered EEG or death compared to infants in the lowest 3 quartiles.

Comments:

1) Methods: Although there have been multiple publications from this trial which provide details of the study, it would be helpful for readers not familiar with the trial to provide a brief description of the study within the methods section (more than currently provided).

2) Methods: Since the results for the CUS are reported, there should be a description of mild, moderate and severe brain injury.

3) Methods/neurodevelopmental assessments: Were examiners blinded to treatment intervention and other clinical data such as imaging results?

4) Methods: definition of neurodevelopmental impairment and CP belong in the methods and not the legend of a table.

5) Methods/statistics: The sentence "If baseline patient characteristics, the interventions conducted during the 72 hours of NIRS... is a run on sentence and is awkward. Although it is recognized that there was no adjustment for neonatal complications, it is not clear why one would consider doing any adjustment for neonatal complications since they may be in the causal path to the outcome at 2 years.

6) Methods/statistics: The two most important issues are as follows;

a. Death is not accounted for in the analyses. Shouldn't the primary outcome be death or a measure of neurodevelopment and a secondary outcome of neurodevelopment among survivors as presented? A value below the lowest measured values for cognitive and motor functions could be assigned to allow an assessment of death for this analysis.

b. There is no adjustment for baseline differences in characteristics among the lowest 3 quartiles and the highest quartile of infants (eg, p values < 0.2 or some other cut point for univariate analyses).

7) Results: the list of baseline characteristics is quite short. Is this the extent of data that is available? Is there information regarding maternal health (diabetes, hypertension), maternal education or sociodemographic status, placental problems (abruption, previa), mode of delivery, histological chorioamnionitis, intubation at birth?

8) Results: umbilical pH is not a treatment.

9) Results/treatment in the first 72 hours; is their information on pneumothorax and placement of a chest tube?

10) Results: the statement "neonatal complication rates were similar" needs more detail as to what complications are being referred to.

11) Results: "The rates of developmental impairment tended to be increased..." Reporting of trends can be problematic and sometimes misleading. It may be more objective to state that the point estimate for outcome x was in the direction of an increase in impairment but the confidence limits were very wide.

12) Discussion/1st paragraph: Discussing trends can be problematic. Consider a more objective statement that the wide confidence limits precludes the ability to assert that there is or is not an association between 2 year outcome and cerebral hypoxia.

13) Discussion: the statement "The fact that the difference in the cerebral outcomes waned with age underline the multifactorial influence in long term outcome..." is an important statement and could be viewed as the conclusion. The authors need to do a better job of integrating this possible conclusion with the limitation of their data. For example as the authors note, although the difference in Bayley scores is approximately a third of a standard deviation in favor of less cerebral hypoxia, the confidence limits for neurodevelopmental impairment are wide. This study is relatively small for 2 year outcome and as noted in the limitations may be under powered. Better integration of these comments would help the message. It seems like it is an open question.

14) Discussion: not performing adjustment of baseline characteristics because there were no differences in the outcome seems very unconventional.

15) Conclusion: This statement should reflect that this was an exploratory analysis. In addition stating that cerebral hypoxia was not statistically associated with 2 year outcome may be misconstrued as more definitive than it is. The wide confidence limits limit the ability to draw conclusions and further study is needed seems like a better way to summarize this interesting data.

Reviewer #2: - This manuscript should include a flow diagram showing how the children recruited to the overarching trial whittle down to the number included in this analysis.

- In the methods, a description of the various tools used for neurodevelopmental evaluation should be included.

- The statistical analysis requires a number of improvements

(1) the table of descriptive characteristics of should be separated from the table describing associations between the main exposure and outcomes. Descriptive characteristics should be summarised in terms of counts and percentages for categorical ones and either means and SDs or medians and IQRs or ranges, for continuous ones. Statistical tests comparing groups are not necessary at this stage, unless the authors have some a-priori hypotheses to test - otherwise these tests simply constitute data dredging/multiple testing (which would be difficult to adjust for in this underpowered study).

(2) a separate table presenting the associations between the main exposure and the various outcomes should be presented. For each outcome either counts (%) or means (SEs) should be presented for each group, followed by the effect estimate, its 95% confidence intervals and p-values. p-values should not be included for subgroup analyses, e.g. the various components of the neurodevelopmental outcome scores, as this also constitutes multiple testing.

Although the authors have previously described the main exposure by contrasting the lower three quartiles of burden of cerebral hypoxia to the uppermost, this approach collapses potentially heterogeneous individuals into fewer groups resulting in loss of information - this is particularly critical in this study where there are few individuals and a small number of outcome events. The authors should include sensitivity analyses where the main exposure is treated as a continuous outcome, to explore linear (or log-linear) trends and departures from linearity/log-linearity in the association between burden of hypoxia and each of the outcomes.

Additionally, more careful consideration should be made about how potential confounders are identified and controlled for. Only potential confounders which could plausibly herald hypoxia (and are not on a causal pathway in the association between the main exposure and each outcome) should be further explored for adjustment.

The methods, results and discussion should then be updated to reflect these changes.

6. PLOS authors have the option to publish the peer review history of their article (what does this mean?). If published, this will include your full peer review and any attached files.

Reviewer #1: No

Reviewer #2: No

---

## [Author Response · Author response to Decision Letter 0]

25 Jan 2021

Georg M. Schmölzer

Academic Editor

PLOS ONE

---

## [Decision Letter · Decision Letter 1]

17 Mar 2021

PONE-D-20-27390R1

Early cerebral hypoxia in extremely preterm infants and neurodevelopmental impairment at 2 year of age: A Post hoc analysis of the SafeBoosC II trial

PLOS ONE

Dear Dr. Anne Mette Plomgaard,

Thank you for submitting your manuscript to PLOS ONE. After careful consideration, we feel that it has merit but does not fully meet PLOS ONE’s publication criteria as it currently stands. Therefore, we invite you to submit a revised version of the manuscript that addresses the points raised during the review process.

We look forward to receiving your revised manuscript.

Kind regards,

Georg M. Schmölzer

Academic Editor

PLOS ONE

Reviewers' comments:

Reviewer's Responses to Questions

**Comments to the Author**

1. If the authors have adequately addressed your comments raised in a previous round of review and you feel that this manuscript is now acceptable for publication, you may indicate that here to bypass the “Comments to the Author” section, enter your conflict of interest statement in the “Confidential to Editor” section, and submit your "Accept" recommendation.

Reviewer #1: All comments have been addressed

Reviewer #2: (No Response)

2. Is the manuscript technically sound, and do the data support the conclusions?

Reviewer #1: Yes

Reviewer #2: Partly

3. Has the statistical analysis been performed appropriately and rigorously? 

Reviewer #1: I Don't Know

Reviewer #2: No

4. Have the authors made all data underlying the findings in their manuscript fully available?

Reviewer #1: Yes

Reviewer #2: No

5. Is the manuscript presented in an intelligible fashion and written in standard English?

Reviewer #1: Yes

Reviewer #2: Yes

6. Review Comments to the Author

Reviewer #1: (No Response)

Reviewer #2: Major comments

I find this manuscript a bit more confusing since the last round of reviews. My understanding is that the authors are exploring the association between cerebral hypoxia (the main exposure) and 2-year outcomes including neurodevelopmental scores, growth, vision and auditory functions, cerebral palsy, etc.

The reason for my confusion is that the approach to the analysis is not clearly articulated and seems incorrect in various places, therefore I am unable to make any sense of the results.

In particular, it is confusing why comparisons such as that in Table 2 are presented: if the hypothesis is that burden of hypoxia might affect outcomes, then performing a statistical test to check whether there are differences in neonatal morbidities and treatments between the groups does not serve any purpose, especially when those morbidities and treatments could influence the outcomes of interest here i.e. these morbidities are on the causal pathway between the main exposure and the outcome. I feel that the authors have not clearly considered the potential relationships (and the directions of those potential relationships) between the factors explored, therefore some of the analyses conducted don't really serve a useful purpose.

Another bit of analysis that doesn't make sense is in Table 3. For example, odds ratios are presented for mental development index and ASQ scores - these are continuous outcome measures for which I would have expected to see mean difference in these measures between the two cerebral hypoxia groups. Presenting odds ratios implies that the authors have explored the effects of the neurodevelopment scores as exposures on hypoxia as the outcome - which is clearly the wrong direction of potential causal relationship between the factors, since the hypoxia was assessed 15 minutes after birth and the nourodevelopmental outcomes at 2 years; therefore the hypoxia is the exposure and the neurodevelopmental score the outcome in this relationship.

Here is what I would suggest the authors do to focus this analysis more clearly:

- make very clear in the methods what the main exposure and outcome are

- the table of descriptive characteristics should present the participant characteristics (already present in table 1), treatments received, and morbidities in the two exposure groups. There is no need to conduct statistical tests comparing these groups in terms of these factors unless some factors are identified which are associated with the main exposure and the main outcomes and not on the causal pathway between them - the problem with the current analysis is that some of these factors are potentially on the causal pathway (for example, the burden of cerebral hypoxia could influence treatment received or neontatal morbidity which could then influence outcomes). Being on the causal pathway means even if the factor is associated with exposure, it is inappropriate to consider adjusting for it in the analysis of the outcome.

- the table of outcomes should list the main outcomes, the means and standard errors of the continuous ones or the counts and proportions/percentages of the binary/categorical ones in each of the hypoxia groups, followed by the estimate of association e.g. mean difference, 95%CI and p-value for continuous outcomes and ORs or RRs with 95%CI and p-values (in that order) for binary outcomes. This analysis should be limited to the main outcomes only and not their components as this would result in multiple testing which there isn't enough power in this study to adjust for.

Minor comments

- the statement just above the 'statistics' section relating to figure 1 belongs to the results section.

- the denominators for each column in Table 2 are missing; there's only 'n=' at the top of each column.

- as advised previously, please report the standard errors (SEs) not standard deviations (SDs) for continuous outcomes in Table 3. SEs are preferred for inference while SDs are appropriately used for description in Table 1.

- also, please report the p-value after the effect estimate and its 95%CI in Table 3.

- the statistical methods say you compared the baseline patient characteristics; in fact, you presented the baseline characteristics, which is appropriate to do i.e. no comparison required, and then you compared the interventions, complications and outcomes between the two groups. Please clarify these in the methods.

7. PLOS authors have the option to publish the peer review history of their article (what does this mean?). If published, this will include your full peer review and any attached files.

Reviewer #1: No

Reviewer #2: No

---

## [Author Response · Author response to Decision Letter 1]

14 Jun 2021

Response to reviewers is attached as one of the files.

---

## [Decision Letter · Decision Letter 2]

9 Jul 2021

PONE-D-20-27390R2

Early cerebral hypoxia in extremely preterm infants and neurodevelopmental impairment at 2 year of age: A Post hoc analysis of the SafeBoosC II trial

PLOS ONE

Dear Dr. Anne Mette Plomgaard,

Thank you for submitting your manuscript to PLOS ONE. After careful consideration, we feel that it has merit but does not fully meet PLOS ONE’s publication criteria as it currently stands. Therefore, we invite you to submit a revised version of the manuscript that addresses the points raised during the review process.

We look forward to receiving your revised manuscript.

Kind regards,

Georg M. Schmölzer

Academic Editor

PLOS ONE

Journal Requirements:

Reviewers' comments:

Reviewer's Responses to Questions

**Comments to the Author**

1. If the authors have adequately addressed your comments raised in a previous round of review and you feel that this manuscript is now acceptable for publication, you may indicate that here to bypass the “Comments to the Author” section, enter your conflict of interest statement in the “Confidential to Editor” section, and submit your "Accept" recommendation.

Reviewer #2: (No Response)

2. Is the manuscript technically sound, and do the data support the conclusions?

Reviewer #2: Partly

3. Has the statistical analysis been performed appropriately and rigorously? 

Reviewer #2: Yes

4. Have the authors made all data underlying the findings in their manuscript fully available?

Reviewer #2: No

5. Is the manuscript presented in an intelligible fashion and written in standard English?

Reviewer #2: Yes

6. Review Comments to the Author

Reviewer #2: - the authors' response regarding the reporting of SDs instead of SEs in Table 3 is unsatisfactory. The authors say that "it is more conventional to give SDs to serve the interpretation of differences on a population scale" - this is incorrect. SDs are descriptive of a sample only. The equivalent statistic that applies to the population is the SE. Please see Altman & Bland's brief discussion of this at https://www.ncbi.nlm.nih.gov/pmc/articles/PMC1255808/. Furthermore, their response that "the p-values also serve to support the interpretation as regards inference" is only partly correct, as p-values are used to make inference with regards to the null hypothesis only; on their own they do not give any information on the uncertainty about estimates of effect or association, which the SEs provide. It remains my recommendation that the SDs in Table 3 and in the text pertaining to the outcomes should be replaced with SEs as is the standard practice in reporting estimates of effect or association for inference.

- all tables need one more improvement: wherever proportions are reported (e.g. proportion of male sex in table 1), these need to be accompanied by counts i.e. in this example, the number of males. Similarly wherever counts are reported e.g. in table 3, the counts for cerebral palsy, these need to be accompanied by the proportions.

- there is still need for some improvement in how the study is described; for example, in the first paragraph on page 4, I needed several reads to understand that the sentence that begins with 'Control group,' was a section describing the control group.

- aspects of the reporting of results also still need improvement; wherever comparing differences between the hypoxia groups, this needs to be clear. For example, statements such as "there were no differences between the Bayley... scores" in the abstract and results sections need to be explicit that this comparison was between the hypoxia groups.

7. PLOS authors have the option to publish the peer review history of their article (what does this mean?). If published, this will include your full peer review and any attached files.

Reviewer #2: No

---

## [Author Response · Author response to Decision Letter 2]

29 Aug 2021

All comments has been addressed in response to reviewers

---

## [Editor Report · Decision Letter 3]

3 Jan 2022

Early cerebral hypoxia in extremely preterm infants and neurodevelopmental impairment at 2 year of age: A Post hoc analysis of the SafeBoosC II trial

PONE-D-20-27390R3

Dear Dr. Anne Mette Plomgaard,

We’re pleased to inform you that your manuscript has been judged scientifically suitable for publication and will be formally accepted for publication once it meets all outstanding technical requirements.

Kind regards,

Georg M. Schmölzer

Academic Editor

PLOS ONE
---

## [Editor Report · Acceptance letter]

13 Jan 2022

PONE-D-20-27390R3 

Early cerebral hypoxia in extremely preterm infants and neurodevelopmental impairment at 2 year of age: A Post hoc analysis of the SafeBoosC II trial 

Dear Dr. Plomgaard:

I'm pleased to inform you that your manuscript has been deemed suitable for publication in PLOS ONE. Congratulations! Your manuscript is now with our production department. 

Kind regards, 

on behalf of

Dr. Georg M. Schmölzer 

%CORR_ED_EDITOR_ROLE%

PLOS ONE